# Ferroelectricity and Oxide Reliability of Stacked Hafnium–Zirconium Oxide Devices

**DOI:** 10.3390/ma16093306

**Published:** 2023-04-23

**Authors:** Ruo-Yin Liao, Hsuan-Han Chen, Ping-Yu Lin, Ting-An Liang, Kuan-Hung Su, I-Cheng Lin, Chen-Hao Wen, Wu-Ching Chou, Hsiao-Hsuan Hsu, Chun-Hu Cheng

**Affiliations:** 1Department of Electro-Physics, National Yang Ming Chiao Tung University, Hsinchu 30010, Taiwan; zoe051285.sc09@nycu.edu.tw (R.-Y.L.); wahaha2900@gmail.com (H.-H.C.); wcchou957@nycu.edu.tw (W.-C.C.); 2Institute of Materials Science and Engineering, National Taipei University of Technology, Taipei 10608, Taiwan; starlin527@gmail.com (P.-Y.L.); howmas13@gmail.com (T.-A.L.); alien79412@hotmail.com (K.-H.S.); allan442211@gmail.com (I.-C.L.); 3Department of Mechatronic Engineering, National Taiwan Normal University, Taipei 10610, Taiwan; 61073021h@gapps.ntnu.edu.tw

**Keywords:** ferroelectric, HfZrO, hysteresis loop, reliability

## Abstract

In this work, we investigate the ferroelectricity of stacked zirconium oxide and hafnium oxide (stacked HfZrO) with different thickness ratios under metal gate stress and simultaneously evaluate the electrical reliability of stacked ferroelectric films. Based on experimental results, we find that the stacked HfZrO films not only exhibited excellent ferroelectricity but also demonstrated a high performance on reliability. The optimized condition of the 45% Zr proportion exhibited a robust ferroelectric polarization value of 32.57 μC/cm^2^, and a polarization current with a peak value of 159.98 μA. Besides this, the ferroelectric stacked HfZrO also demonstrated good reliability with a ten-year lifetime under >−2 V constant voltage stress. Therefore, the appropriate modulation of zirconium proportion in stacked HfZrO showed great promise for integrating in high-performance ferroelectric memory.

## 1. Introduction

With the trend of technological development, concepts such as the Internet of Things (IoT), artificial intelligence (AI), 5G (The fifth generation of mobile internet connectivity) communication, and cloud computing are rapidly emerging, and the metaverse has attracted much attention in recent years. Therefore, the demand for memory, which plays a core role, has gradually increased. The requirements of the memory market look to bo developing exponentially. Since the first half of 2020, a new type of working mode has progressively emerged. In view of the change in living and working habits, the stay-at-home economy and working-from-home have also driven the growth of notebook computers and mobile communications, resulting in a sharp increase in demand for chips. Ferroelectric materials have been investigated for several decades [1]. Hafnium-oxide (HfO_2_)-based ferroelectric materials are often investigated and proposed to replace the traditional ferroelectric perovskites, such as PbZr*_x_*Ti_1–*x*_O_3_ and BaTiO_3_ [2,3,4]. Traditional perovskite ferroelectric materials have some issues, such as poor scalability, being difficult to etch, and causing environmental pollution of lead contamination, which affects the development of ferroelectric materials and devices for the applications of low-power advanced memory and intelligent computing. Compared with traditional ferroelectric materials, HfO_2_-based ferroelectric materials have better remanent polarization and coercive electric field. Moreover, compared with traditional ferroelectric materials, HfO_2_-based ferroelectric materials are easier to conform with the surface during deposition, and are also compatible with standard semiconductor device technology. Thus, the production of HfO_2_-based ferroelectric films using atomic layer deposition has already matured. To develop the technology, HfO_2_-based ferroelectric materials with thickness scaling have been investigated based on complementary metal oxide semiconductor (CMOS) technology for logic and memory device applications [5,6,7,8], due to their inherent advantages, such as low operation voltage and CMOS compatibility. These advantages allow them to overcome unfavorable obstacles originating from traditional perovskite ferroelectric material. Therefore, it is considered a promising candidate for ferroelectric memory and neuromorphic computing applications [9,10,11,12,13]. Since the ferroelectric properties of lightly Si-doped HfO_2_-based ferroelectric thin films were first reported in 2011, doped HfO_2_-based ferroelectric materials have been extensively studied. It is generally accepted that ferroelectricity in HfO_2_ can be obtained in a specific dopant concentration range, and its spontaneous polarization is caused by the displacement of oxygen ions in non-centrosymmetric orthorhombic crystals. The mechanical stress applied by the metal electrode and the subsequent thermal annealing are also responsible for the phase transformation of the ferroelectric orthorhombic phase in HfO_2_-based thin films. The related studies also pointed out that mechanical stress formed by the top electrode and additional dopant incorporation into the HfO_2_ film can stabilize the ferroelectric phase and simultaneously inhibit the formation of the non-ferroelectric phase during ferroelectric annealing. For novel HfO_2_-based materials, the formation of a non-centrosymmetric orthorhombic phase with ferroelectric properties can be achieved by many methods, such as doping (Zr, Al, Sr…) [14,15], thickness modulation [16,17], capping layer [18,19,20], metal gate stress engineering [21,22,23,24], and formation of oxygen vacancies [25], etc. The hafnium–zirconium oxide (HfZrO) films with an extensive doping range for the modification of the ferroelectric polarization characteristics have been expansively investigated in recent years [26]. Since the zirconium doping concentration in HfO_2_ films can achieve a maximum ferroelectric polarization strength under a large doping ratio of 50%, it is beneficial for the thickness scaling for high-density memory or storage applications. However, the ferroelectric polarization characteristic gradually decays while the doping concentration of zirconium increases more than 50%. The high content of zirconium doping easily leads to a weak film quality, especially at high temperatures, which may cause serious leakage issues [27]. Previous studies have claimed that the poor thermal stability of zirconium oxide within the HfZrO might cause severe thermal reactions in the interface, which creates excessive defects and induces an additional leakage path during high-temperature annealing. In order to improve these problems, the stacked structure of HfZrO films has been proposed. From our previous work, the leakage current can be effectively suppressed in stacked HfZrO films, and the excellent ferroelectric polarization characteristics also can be maintained [28]. According to our previous experimental results, it can be concluded that the stacked HfZrO ferroelectric capacitor has relatively low leakage current and good thermal stability under PMA processes than the mixed HfZrO ferroelectric capacitor. This is because the leakage current path caused by the diffusion of ZrO_2_ near the Si interface can be effectively interrupted by the buffer layer HfO_2_ in the stacked film structure of ZrO_2_/HfO_2_. In contrast, the leakage path may not be easy to control in mixed HfZrO thin films because the diffusion coefficient of zirconium atoms is much higher than that of hafnium atoms. Therefore, the stacked structure is utilized in this study to diminish the influence of poor thermal stability caused by the unstable Zr dopant. To consummately appraise the ferroelectricity of stacked HfZrO films, we not only describe the ferroelectric hysteresis behavior, and the ferroelectric crystallinity but also discuss the leakage current characteristics and stress reliability. In addition, we calculate the ten-year lifetime and trapping level in this article.

## 2. Materials and Methods

In order to figure out the ferroelectric properties of the stacked HfZrO films, we made the ferroelectric capacitors with different zirconium proportions. The stacked HfZrO ferroelectric capacitors were fabricated on the highly doped n+-type silicon substrate with an arsenic doping concentration of 1.24 × 10^19^ cm^−3^ to characterize the ferroelectric properties of thin films. First, the silicon substrate was followed by Radio Corporation of America (RCA) clean to remove particles and native oxide. A 1 nm chemical oxide was grown on n+ substrate by pure H_2_O_2_ solution as a buffer layer. Then, 10 nm thick stacked HfZrO films were fabricated by means of atomic layer deposition (ALD) at 250 °C. Tetrakis(dimethylamino)zirconium (Zr[N(CH_3_)_2_]_4_) and Tetrakis(dimethylamino)hafnium (Hf[N(CH_3_)_2_]_4_) were employed as the precursors for zirconium and hafnium, respectively. Subsequently, the ferroelectric capacitors were patterned through optical lithography. After that, the 200 nm thick tantalum nitride (TaN) was deposited using the DC sputtering system as the top electrode of the ferroelectric structures. The argon and nitrogen flow rates in the sputtering system were 100 and 10 sccm, respectively, during the deposition of TaN top electrode. Finally, the stacked HfZrO films was annealed in N_2_ ambient for 30 s to crystallization by post-metallization annealing (PMA) at the optimized temperature of 500 °C, which was applied to induce phase transition to the ferroelectric crystal phase by metal gate strain engineering. The high-resolution transmission electron microscopy (HR-TEM) using a JEOL JEM-2010F at 200 kV was employed for microstructure observation of device structure. Figure 1a–c shows the TEM cross-sectional images of crystallized stacked HfZrO_x_ films with 30%, 45%, and 60% Zr proportions, respectively. The crystalline phases of stacked HfZrO films with different zirconium proportions were measured by grazing incidence X-ray diffraction (GI-XRD) of Bruker D8 discover using Cu K-α radiation (the wavelength is about 1.54 Å). To avoid the influence of substrate peaks, a low incidence angle of 1° was applied. The ferroelectric polarization characteristics were measured by a Precision RT66C ferroelectric tester (Radiant Technology, Inc.). The Positive Up Negative Down measurements were performed by Agilent B1500A semiconductor analyzer and B1530A waveform generator.

## 3. Results and Discussion

Figure 2a shows the hysteresis loop of stacked HfZrO ferroelectric capacitors with various Zr proportions under an applied voltage of 4 V. Before the measurement of the hysteresis loop, the wake-up process was performed for 10^3^ operating cycles at an operating voltage of ±4 V. The purpose is to prevent the ferroelectric dipole from being affected by the pinning of oxygen vacancies [29] and achieve obvious ferroelectric characteristics. In this work, there is no apparent change in remanent polarization (2P_r_) after the wake-up process. The values of 2P_r_ corresponding to the Zr proportions of 30%, 45%, and 60% are 17.3 μC/cm^2^, 32.57 μC/cm^2^, and 21.25 μC/cm^2^, respectively. It can be clearly observed that the sample of 30% Zr proportion presents the weakest ferroelectric property of all. When the proportion of Zr increases to 45%, the ferroelectric properties of the device becomes dramatically enhanced. The ferroelectric characteristic of the instantaneous current was also measured, as shown in Figure 2b. Compared to others, the instantaneous current of the HfZrO film with a 45% Zr proportion was correspondingly more pronounced, which is consistent with the measured result of Figure 2a. In order to further observe the ferroelectric characteristics of stacked HfO_2_/ZrO_2_ films with different Zr proportions, we adopted the Positive Up Negative Down (PUND) measurement method to extract polarization current (I_P_) to confirm the ferroelectric characteristic. As shown in Figure 3a, the pulse condition is an input voltage of ±3.5 V with a pulse width of 50 μs and a raising of 10 μs. The waveform contains two positive pulses, two negative pulses, and a preset pulse. By subtracting the non-switching pulse from the switching pulse, the leakage contribution to the polarization can be eliminated, thereby confirming the true polarization of the device. The ferroelectric polarization current can be extracted from the PUND measurement, excluding the contribution of the dielectric and leakage current constituents [30]. Figure 3b shows the polarization currents extracted by PUND measurement. The result indicates that the stacked HfZrO film with a 45% Zr proportion has stronger ferroelectricity, which involves the largest polarization current with a peak value of 159.98 μA. The HfZrO film with a 30% Zr proportion shows a comparatively low polarization current of 43.74 μA, which is approximately four times lower than that of a 45% Zr proportion. Besides, it can be observed that the degradation of polarization current with increasing Zr proportion up to 60%. It can be explained by the formation of a non-ferroelectric tetragonal phase transition in staked HfZrO with a higher Zr proportion.

Figure 4 shows the GI-XRD spectrum of 10 nm thick stacked HfZrO films with various proportions of zirconium. The GI-XRD result indicates the presence of orthorhombic crystal phase (o-phase) in stacked HfZrO; it shows a good accordance to the o-phase at 30.2° (111) and 35.5° (002). It is well known that the ferroelectricity of HfZrO films originates from the formation of the o-phase [31]. The GI-XRD pattern of the sample with a 30% Zr proportion shows a slight intensity of peak to the monoclinic phase (m-phase) at 28.3° (−111), and the sample with a 60% Zr proportion shows a slight intensity of peaks to the tetragonal phase (t-phase) at 49.5° (112) and 59.4° (211), whereas that of the sample with 45% Zr presents the strongest diffraction peaks at 30.2° (111) with the most intense o-phase. Therefore, we can confirm that the strong ferroelectricity observed in the sample of 45% Zr mainly originates from the increase in the ferroelectric crystalline phase.

It is known from previous studies that the metal gate can provide mechanical stress to induce the generation of the ferroelectric o-phase in the PMA process. To further evaluate the influence of the metal gate stress effect on stacked HfZrO films, the strain fields of HfZrO capacitors were analyzed by geometrical phase analysis (GPA) to investigate the mechanical stress at gate stacks. The ε_xx_ and ε_yy_ strain fields of stacked HfZrO films with 30%, 45%, and 60% Zr proportions were extracted from the atomic displacements of TEM cross-sectional images, as shown in Figure 5a–c. The ε_xx_ and ε_yy_ represent the strain along the direction of the x axis in the x plane and the direction of the y axis in the y plane, respectively. We can observe that the intensity of the strain field ε_yy_ in stacked HfZrO films is stronger than the strain field ε_xx_. Since the y axis direction is along the film growth direction (normal to the surface of silicon substrate), the tensile stress and tensile strain along the y axis direction possibly affect the transition of the o-phase during the PMA process. Therefore, the ferroelectric phase transition is correlated with the thermal stress caused by the thermal expansion coefficient difference between TaN metal electrode and ferroelectric HfZrO thin film under the PMA process. It follows that the proper control of thermal stress can maintain the stability of the metastable non-centrosymmetric o-phase and ferroelectric polarization properties of HfZrO.

It is well known that the conduction mechanism of the leakage current is apparently influenced by electric field and temperature. In order to study the leakage current mechanism under high field conditions, this section will discuss the variation of leakage current through variable temperature and trapping level of the stacked HfZrO films. Figure 6a–c shows the leakage currents measured at room temperature (25 °C) and high temperature (50 °C, 75 °C, 100 °C, 125 °C) of stacked HfZrO films with Zr proportions of 30%, 45%, and 60%. It can be found that the gate leakage current apparently becomes large with increasing the temperature from 25 °C to 125 °C. Furthermore, we also observed that the high-temperature leakage current at negative bias significantly increases, especially for the case of higher 60% Zr proportion. This can be attributed to the high-temperature-induced leakage current caused by interface or bulk defect sites. On the other hand, to further obtain the trapping levels of the stacked HfZrO films, the current–voltage relationship of the stacked HfZrO films at different temperatures was converted into an Arrhenius diagram and calculated, as shown in Figure 7a–c. The average trapping level for the Zr proportions of 30%, 45%, and 60% ware 0.4358 eV, 0.3932 eV, and 0.3195 eV, respectively. Overall, the trapping level in the stacked films is constituted by defects, such as impurities, vacancies, and grain boundaries. It can be observed that the smaller trapping level occurred at higher Zr proportions, leading to a smaller trapping level. The change in trapping level also agrees with the XRD results, showing the decrease in the full width at half the maximum height of the o-phase (111) peak while increasing Zr proportion from 30% to 60%. This is because the trap energy level of ZrO_2_ is shallower than HfO_2_, and it is easier to create defects at the interface [32].

To further study the failure mechanism in the gate dielectric, the time-dependent dielectric breakdown (TDDB) method with the constant voltage stress (CVS) test was utilized, which has been proposed for predicting the lifetime of dielectric layers. In view of the above results, the stacked HfZrO capacitors with the Zr proportions of 30%, 45%, and 60% were measured using the CVS. From the relationship between stress time and leakage current, the time for a dielectric breakdown of the device under different constant voltage measurements can be summarized. Figure 8a–c shows the results of the CVS conditions under applied voltage at −3.5 V, −4 V and, −4.5 V of a stacked HfZrO film with the Zr proportions of 30%, 45%, and 60%, respectively. We can observe the longer stress time of >4 × 10^4^ s obtained for the case of 45% Zr proportion at a low-CVS condition of −3.5 V. When the CVS is performed under a large voltage of −4.5 V, the hole generation can be clearly observed and the hard breakdown occurs. The stress time for reaching dielectric hard breakdown is about 10~100 s under −4.5 V for all these three cases.

Figure 9 shows the time-dependent ferroelectric breakdown of stacked HfZrO capacitors with the Zr proportions of 30%, 45%, and 60%. Based on the linear fitting from the current–time plot, the voltage that can allow the device to operate for ten years can be inferred. The extrapolated voltages at a ten-year lifetime for Zr proportion of 30%, 45%, and 60% are −1.32 V, −2.21 V, and −0.96 V, respectively. It is significant that the case of 45% Zr has the largest ten-year extrapolated operating voltage, which was consistent with the previous results of electrical characteristics. Therefore, from the electrical reliability results, we can confirm that the thickness scaling of ZrO_2_ is beneficial for the reliability of stacked HfZrO films.

## 4. Conclusions

In this work, we successfully demonstrated the ferroelectric polarization characteristics of stacked HfZrO ferroelectric films. According to the experimental results, it was confirmed that the metal gate stress would transfer to the stacked HfZrO films, which significantly accelerates the ferroelectric o-phase and stabilizes the polarization effect. Furthermore, the sample with a 45% Zr proportion features an outstanding ferroelectric property and reliability, which was predominantly implemented by metal gate stress and well-controlled Zr proportion calibration. To sum up, an appropriate proportion of Zr/Hf ratio in a HfO_2_/ZrO_2_ gate stack structure with metal gate engineering can certainly realize the high performance of reliability required for emerging memory technology applications.

## Figures and Tables

**Figure 1 materials-16-03306-f001:**
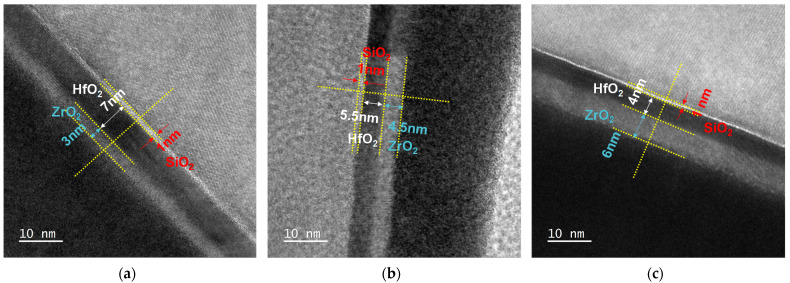
Cross-sectional HRTEM images of (**a**) 30% (**b**) 45%, and (**c**) 60% Zr proportion stacked HfZrO thin films.

**Figure 2 materials-16-03306-f002:**
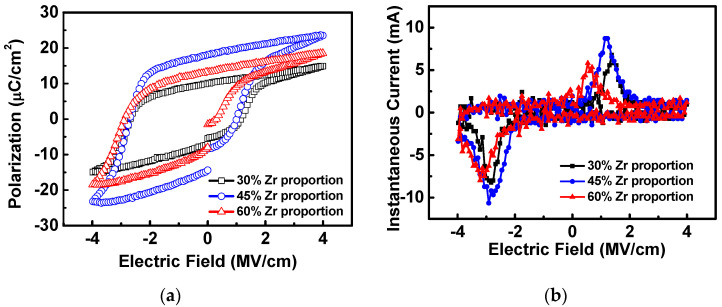
(**a**) Hysteresis loops and (**b**) instantaneous currents of the stacked HfZrO thin films with various Zr proportions.

**Figure 3 materials-16-03306-f003:**
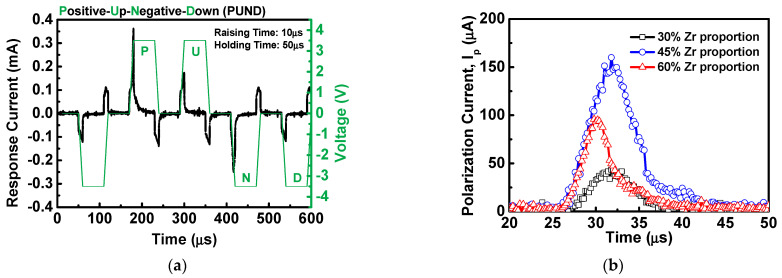
(**a**) PUND measurement waveform diagram and (**b**) polarization currents of stacked HfZrO films with different Zr proportions.

**Figure 4 materials-16-03306-f004:**
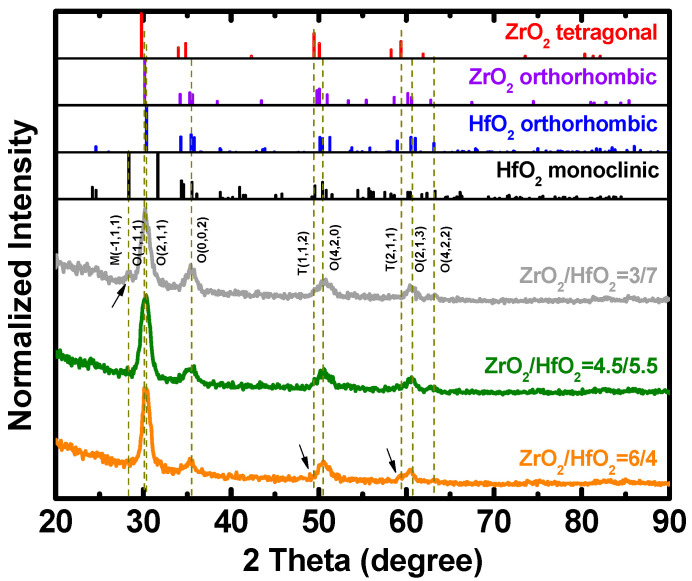
GI-XRD spectrum of stacked HfZrO films.

**Figure 5 materials-16-03306-f005:**
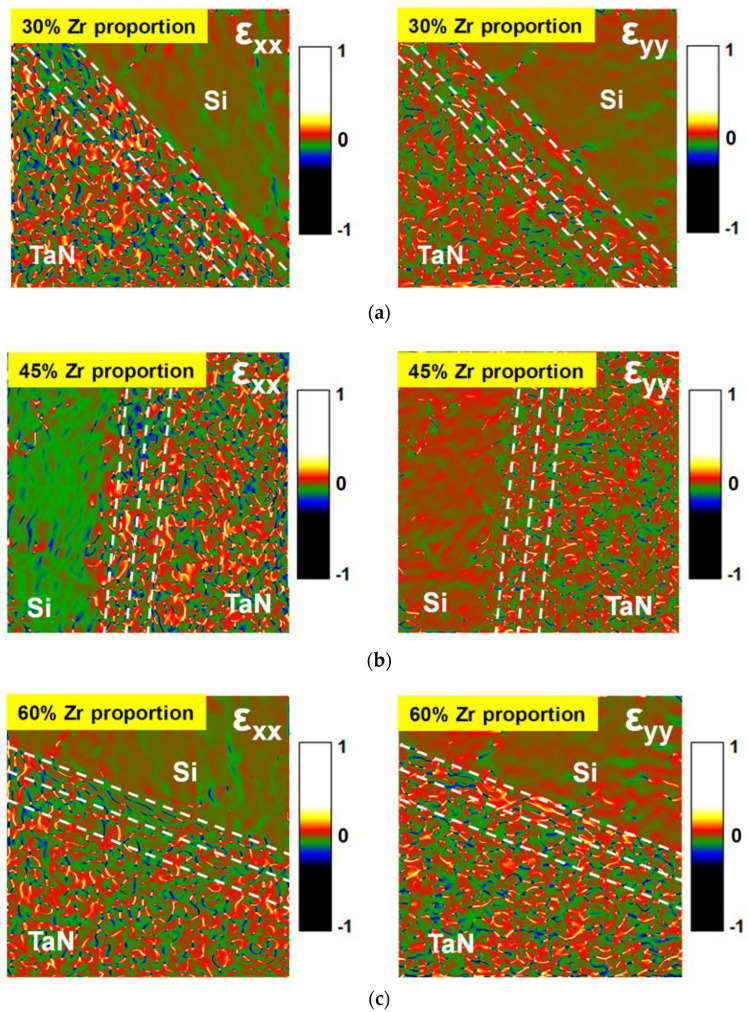
ε_xx_ and ε_yy_ strain fields of stacked HfZrO films with (**a**) 30%, (**b**) 45%, and (**c**) 60% proportion of Zr.

**Figure 6 materials-16-03306-f006:**
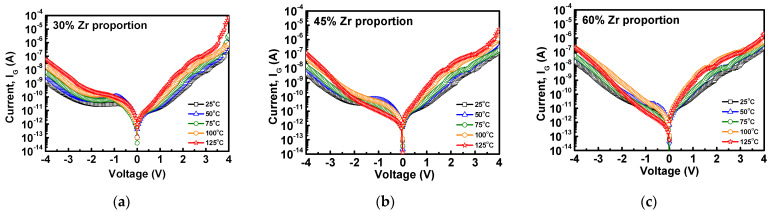
Temperature dependence of leakage current of (**a**) 30%, (**b**) 45%, and (**c**) 60% Zr proportion in stacked HfZrO films.

**Figure 7 materials-16-03306-f007:**
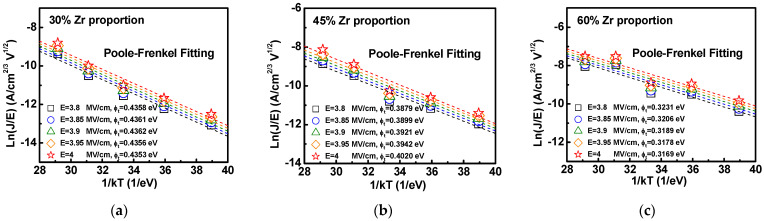
Trapping levels obtained by Poole–Frenkel fitting of (**a**) 30%, (**b**) 45%, and (**c**) 60% Zr proportion in stacked HfZrO films.

**Figure 8 materials-16-03306-f008:**
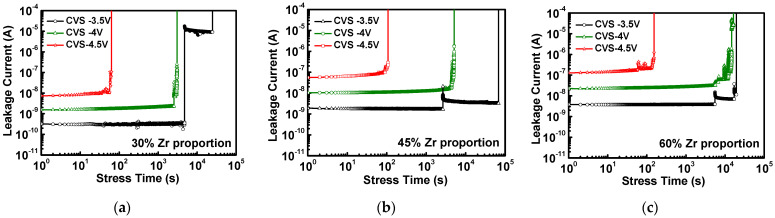
Current–time plots of ferroelectric stacked HfZrO films of (**a**) 30%, (**b**) 45%, and (**c**) 60% Zr proportion under CVS.

**Figure 9 materials-16-03306-f009:**
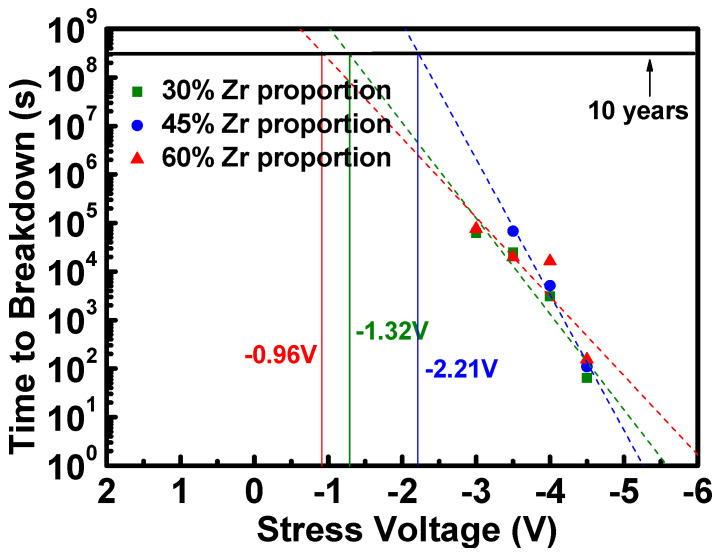
Ten-year lifetime comparison of ferroelectric stacked HfZrO films of (**a**) 30%, (**b**) 45%, and (**c**) 60% proportion of Zr.

## Data Availability

Not applicable.

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
