# Peer review of "Ferroelectricity and Oxide Reliability of Stacked Hafnium–Zirconium Oxide Devices"

_materials, 2023, doi:10.3390/ma16093306_

Round 1

Reviewer 1 Report

The Article is devoted to the search for opportunities to improve the ferroelectric and dielectric properties of materials that are a stacked structure of HfZrO films. A comparison of the ferroelectric characteristics and leakage current in stacked films showed that the best parameters are realized in a system with a ratio of Zr proportion of 45%.

The Article is of interest for Materials and may be published after small revision.

The Article contains stylistic errors. Some are shown in the comments in the attached pdf file. According to the reviewer, the part about strains and stresses should be written more convincingly. In particular, it is desirable to give numerical characteristics showing which deformations prevail in the studied films.

The Article contains stylistic errors. Some are shown in the comments in the attached pdf file. 

Reviewer 2 Report

The manuscript by Liao et al entitled “Ferroelectricity and Oxide Reliability of Stacked Hafnium-Zirconium-Oxide Devices” describes experimental studies of stacked Zr-doped Hafnium oxide thin films. They fabricated thin films on n+ Silicon substrate by the ALD.  They claimed excellent ferroelectricity and high performance on reliability for the optimized Zr proportion of 45% which showed a polarization maximum of 32.57 μC/cm2 , and the polarization current with a peak value of 20 159.98 μA.

The results are of very good quality and the research will be interesting to the community of Materials journal.

HfO2:Zr is an important ferroelectric member of CMOS and the results get the attention of researchers in this field.

The manuscript is written clearly.

I feel the authors should clarify the following in order to demonstrate the significance of present work.

Visual observation of XRD shows decreased full width at half maximum of peak around 2theta of 30 degrees. This is related to the grain size or increase in boundaries.

This is closely related to the line 168 which mentioned, “Overall, the trapping level in stacked film is 168 constituted by defects, such as impurities, vacancies, and grain boundaries..”

The work on HfO2:Zr has been mentioned in several articles in the literature. What way the present work different from the PVD grown epitaxial oxides? Can they correlate or differentiate domain switching characteristics?

Reviewer 3 Report

Reviewer Comments to Authors

The article is interesting from the application point of view. Below you can find my comments.

1.      Line 62: Please explain the abbreviation RCA

2.      Line 85: Did you use 10 nm thick HfZrO layers for these measurements? I'm interested in what the electric field is when 4V is applied.

3.      Line 85: Can you explain what happens during the "the wake-up process"?

4.      Line 91-97: Using a hysteresis loop to determine the content of the orthorhombic ferroelectric phase is not a good idea. It is known that the shape of the loop depends on defects, near-electrode processes, etc. This is evidenced by the observed asymmetry of the current in Fig. 2b. I recommend referring to structured data.

5.      It is easier for the reader to use one term, while the authors used "ZrO2/HfO2=3/7" in the figures, and "30% Zr" in the text; similarly for other ZrO2 contents. You might consider unifying it.

6.       Line 131-133: The authors stated, "Therefore, we can confirm that the strong ferroelectricity observed in the sample of 45% Zr is mainly originated from the increase of the ferroelectric crystalline phase." Have you considered the effects of phase mixture and resulting stress (not only from metal gate) to increase polarization/ferroelectricity?

7.      The description of the scales in Figure 5 can be improved.

8.      The labels in figure 7 are 1/KT, if K is Boltzmann's constant I recommend using lower case k.

9.      I understand that "a ten-year lifetime" is just a theoretical fit. How many samples did you check for each composition?

10.  How many samples did you check to state "the sample of 45% Zr proportion features an outstanding ferroelectric property and reliability"? What I mean is how repeatable is all this data from the composition point of view.

Reviewer 4 Report

Maybe composite HfZrO films with 50% Zr proportion have better ferroelectric properties than with 45% Zr proportion? What is the physico-chemical basis of the influence of the thickness ratio of ZrO and HfO layers on the dominance of the ferroelectric phase?

Why are the diffraction peaks of Si and TaN not also visible in the XRD spectrum on Figure 4?

Line 72: It seams that the phrase “was applied to induced” shoud be replaced by “was applied to induce”.
